# A nowcasting framework for correcting for reporting delays in malaria surveillance

**Tigist F. Menkir**[1], **Horace Cox**[2], **Canelle Poirier**[3,4], **Melanie Saul**[2], **Sharon Jones-Weekes**[2†], **Collette Clementson**[2], **Pablo M. de Salazar**[1], **Mauricio Santillana**[1,3,4☉]*, **Caroline O. Buckee**[1☉]*

**1** Center for Communicable Disease Dynamics, Department of Epidemiology, Harvard T.H. Chan School of Public Health, Harvard University, Boston, Massachusetts, United States of America, **2** Vector Control Services, Ministry of Public Health, Georgetown, Guyana, **3** Computational Health Informatics Program, Boston Children's Hospital, Boston Massachusetts, United States of America, **4** Department of Pediatrics, Harvard Medical School, Boston, Massachusetts, United States of America

☉ These authors contributed equally to this work.
† Deceased.
* msantill@fas.harvard.edu (MS); cbuckee@hsph.harvard.edu (COB)

**Data Availability Statement:** All input data and code is available at https://github.com/goshgondar2018/guyana_nowcasting.

## Abstract

Time lags in reporting to national surveillance systems represent a major barrier for the control of infectious diseases, preventing timely decision making and resource allocation. This issue is particularly acute for infectious diseases like malaria, which often impact rural and remote communities the hardest. In Guyana, a country located in South America, poor connectivity among remote malaria-endemic regions hampers surveillance efforts, making reporting delays a key challenge for elimination. Here, we analyze 13 years of malaria surveillance data, identifying key correlates of time lags between clinical cases occurring and being added to the central data system. We develop nowcasting methods that use historical patterns of reporting delays to estimate occurred-but-not-reported monthly malaria cases. To assess their performance, we implemented them retrospectively, using only information that would have been available at the time of estimation, and found that they substantially enhanced the estimates of malaria cases. Specifically, we found that the best performing models achieved up to two-fold improvements in accuracy (or error reduction) over known cases in selected regions. Our approach provides a simple, generalizable tool to improve malaria surveillance in endemic countries and is currently being implemented to help guide existing resource allocation and elimination efforts.

## Author summary

Infectious disease monitoring systems worldwide are often threatened by severe reporting delays, hindering resource planning efforts, particularly for endemic diseases like malaria and in remote localities. In Guyana, a country located in South America, regions facing infrastructural barriers and consisting of highly mobile or socially disadvantaged populations oftentimes face the greatest delays in reporting local health facility data to the country's central database. To better characterize and address this issue, we analyzed time and

**Funding:** TFM was supported by the National Institute of General Medical Sciences of the National Institutes of Health [U54GM088558]. PMD was supported by the Fellowship Foundation Ramon Areces. MS was partially supported by the National Institute of General Medical Sciences of the National Institutes of Health [R01 GM130668]. MS and CP acknowledge that they have received institutional research funds from the Johnson and Johnson Foundation. COB was supported by a NIGMS Maximizing Investigator's Research Award (MIRA) [R35GM124715-02]. The content is solely the responsibility of the authors and does not necessarily represent the official views of the National Institutes of Health. The funders had no role in study design, data collection and analysis, decision to publish, or preparation of the manuscript.

**Competing interests:** The authors have declared that no competing interests exist. Author Sharon Jones-Weekes was unable to confirm their authorship contributions. On their behalf, the corresponding author has reported their contributions to the best of their knowledge.

spatial trends in malaria reporting delays across administrative regions and implement methods to transform incomplete case data to more accurate current case counts. We first identify factors which significantly spatially overlap with areas marked by increased reporting delays. We then introduce methods which build on prior case information only available at the time of the target month of interest to anticipate the total number of cases to occur that month, a fraction of which are not known until the subsequent month(s). We compared our regional monthly estimates with eventually reported cases and found that they were up to two times more accurate than case counts known at the month of reporting. Our models provide a widely accessible and adaptable tool for improving on malaria surveillance efforts.

## Introduction

Malaria continues to be one of the most important causes of morbidity and mortality globally—with 405,000 deaths estimated to have occurred in 2018—particularly in poor and rural populations [1]. National malaria surveillance programs depend on timely reporting rates to identify signals of increasing malaria activity and potential outbreaks to ensure that control resources are optimally and most efficiently deployed [2]. However, major barriers to timely surveillance in endemic countries—particularly those in which malaria transmission is concentrated in the most remote, hardest to reach populations—include limited connectivity, modes of digitisation and other context-specific factors that undermine the ability to update surveillance records without significant delays [3]. While infrastructure development is critical to improving the timeliness of case reporting, methodological approaches to adjust for known reporting lags are urgently needed until data systems improve.

One such endemic context is Guyana, located in the northern coast of South America and bordering Venezuela, Suriname, and Brazil; one of 21 malaria endemic countries in the Americas [4]. Guyana has a relatively small, sparse population of ~800,000 people [5], with highly seasonal malaria endemicity concentrated in rainforest regions with low accessibility, often only reachable by small plane or boat. While confirmed cases have steeply declined from 2015 to 2017 in South America overall, Guyana has experienced a near 40% escalation in confirmed cases during that period, with 13,936 cases confirmed in 2017 [4]. More than half of infections are caused by *Plasmodium vivax*; however, the second most common (44%) infecting species, *P. falciparum*, is tied to more severe outcomes and has been implicated in recent reports of resistance to Artemisinin drugs [6–8]. The recent growth in case counts overall is partially driven by increased importation from surrounding areas, such as Venezuela, linked with migration from social and political crises and to promising employment opportunities in mining [4,9]. In response to the elevated burden of malaria in the nation, the government has boosted investment in malaria prevention, under the direction of the Vector Control Services (VCS) [4,10].

Like many national malaria control programs, one of the priority areas of VCS is surveillance, where regional health offices and the central office in the capital of Georgetown engage in a complex surveillance cascade. Cases are largely identified through a passive surveillance system, i.e. patients presenting with symptoms are diagnosed at local health facilities, and rarely through alternative community testing or outreach programs; a mere 8% of all cases are detected through these auxiliary programs. Health facilities in endemic regions are then required to send their daily case registers for all malaria patients to their corresponding regional health office for processing, before being submitted to the central office in the capital

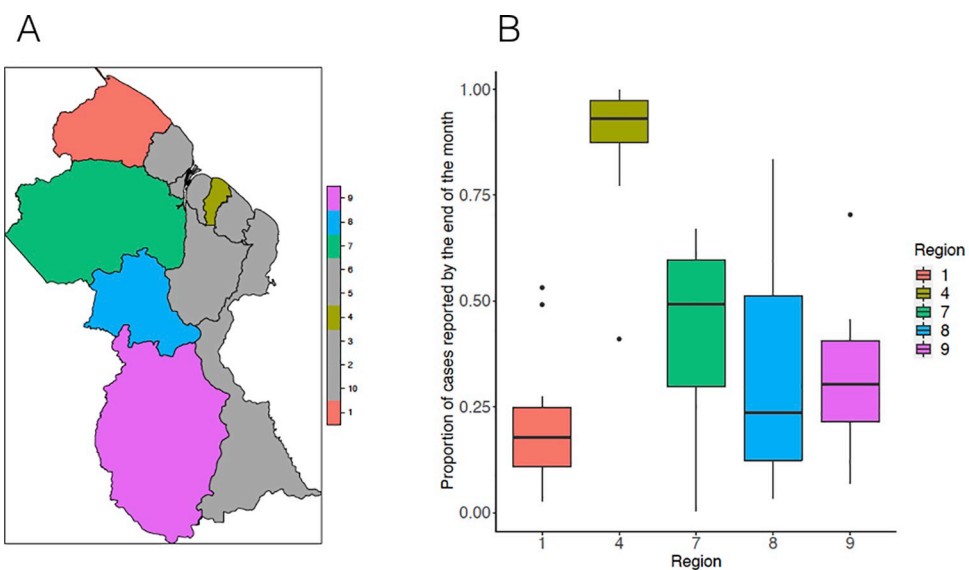

**Fig 1. A. Map of malaria endemic regions and region 4; B. Boxplot reporting the proportion of annually aggregated cases from 2006–2019 reported by the end of the month for each region, with colors corresponding to the regions depicted in the map.** *Map created using the R package ssplot.*

of Georgetown, while health facilities in non-endemic regions can simply send reports to Georgetown [11]. However, we find that many health facilities in endemic areas opt to send their reports directly to Georgetown or through alternative routes, due to issues in road access and other challenges, often resulting in delayed or missed reports (*personal communication*). Furthermore, while the data is intended to be documented in the system in near real-time, with summary reports sent from regions to the central office via USBs, physical copies of individual case registers from facilities are sometimes delivered to the central office, leading to additional bottlenecks in reporting (*personal communication*). Together, these challenges contribute to the number of cases recorded in the central database at the end of the month potentially failing to reflect the true burden of disease at a given time, with cases typically reported more than a month after they occur (See Fig 1B). These issues are not only common for malaria control and surveillance programs, but for many other epidemiological reporting systems, hindering the ability of public health programs to implement appropriate and optimal measures for control [3,12–16]. Modeling efforts to translate available reporting data to more meaningful and accurate measures of real-time incidence could help address some of these challenges.

A number of analytical approaches aimed at improving estimates through "nowcasting" have been developed to account for reporting delays (frequently referred to as "backfill"). The most widely used are Bayesian estimation and regularized regression, which take advantage of the fact that the timing of delays may be relatively predictable. The Bayesian approach, broadly considered the paragon approach, encapsulates both delay trends given prior known cases and epidemiological trends to estimate disease occurrence and mortality, with several recent extensions implemented to better uncover the true magnitude of COVID-19 cases and deaths [14,15,17–23]. Extensions of these methods, proposed by Bastos et al., Rotejanaprasert et al., and Kline et al. additionally accommodate spatial trends in transmission and delay dynamics [14,17,19]. Relatedly, Reich et al. use a generalized additive model to estimate Dengue Hemorrhagic Fever cases in Thailand, modeling incidence at each time point in a given province as a function of its preceding reported cases and those of provinces with similar activity [16].

Penalized regression methods, such as the ARGO, Net, and ARGONet models proposed by Santillana, Lu, and others have also been used to incorporate a diversity of case reporting and alternative data sources, to learn from previous time trends in delays and transmission, and the spatial dependencies in these trends, to predict rates of influenza-like illness [24]. However, this class of models is only well-suited for contexts in which these varied digital sources are readily available and easily integrated into routine model training, testing, and validation efforts [24].

To address challenges in reporting delays for malaria surveillance in Guyana, we developed nowcasting approaches to estimate monthly cases of malaria in each endemic region, using data from the central VCS database from 2006 to 2019 to develop and test our model. We show that proximity to high-risk vulnerable populations, namely mining sites and Amerindian populations, is associated with reporting delays and that delays are relatively consistent over time. Importantly, our predictions provide extensive improvements in surveillance capacity for remote areas and are presently being used by VCS to help characterize prevailing case counts across regions in the face of added challenges due to COVID-19, to help facilitate the allocation of malaria prevention and control interventions and overall planning initiatives centered on elimination. Historically, malaria has affected groups that primarily live and work in Regions 1, 7, 8, and 9, otherwise known as hinterland regions because they are sparsely populated, forested areas that are difficult to navigate. These characteristics contribute to the ongoing challenge of time lags in reporting of malaria cases which threaten the delivery of an appropriate public health response. The nowcasting tool generates estimates of current malaria burden at the sub-national level based on real-time reporting and a contextual understanding of the malaria epi-trends over the past years. Thus, we illustrate how nowcasting methods offer a general, tractable approach for improving decision-making for malaria control programs in countries that have significant reporting delays.

## Methods

### Ethics statement

The Ministry of Public Health, Guyana granted access to non-identifiable surveillance data obtained from the Malaria Program /Vector Control Services under IRB exemption. Further, IRB exemption was obtained from Harvard School of Public Health (US), Human Research Protection Program, Protocol number IRB18-1638.

We had access to daily case register data from 2006 to 2019 documented at health facilities in each of Guyana's ten regions and compiled at the central office in Georgetown (region 4), which consists of demographic information, results from tests concerning the identity of the infecting parasite species, and the dates when the patient's smear was taken, examined, and inputted in the central office. Delays are defined as the days elapsing from when a patient is registered as a case at a health facility and when the patient is documented at the central VCS database. We excluded the 5.7% of observations with implausible recorded date ranges, i.e. individuals whose smear was examined in a health facility following their documentation in the central office in Georgetown. A majority of the 28% of cases indicated as rechecks were noted to have been infected by P. vivax, or mixed infections with P. vivax (64%), as expected, given that P. vivax is the species largely implicated in relapses [25]. While we acknowledge that the inclusion of patients with recurrent infections may hinder a mechanistic analysis of potential transmission patterns in each region, the primary goal of this analysis is to learn from prior clinical reporting trends to arrive at more informed estimates of current case loads, which include all patients, in the face of reporting delays.

We assessed the potential influence of rainfall on reporting delays by analyzing precipitation data sourced from East Anglia University's Climate Research Unit's gridded dataset repository. Monthly total precipitation data was extracted for each region by year [26]. Locations of second-level administrative regions, or neighborhood democratic councils (NDCs) (n = 116) were identified through a shapefile of NDCs from GADM version 1 (accessed through DIVA-GIS) [27]. We defined NDCs, rather than regions, as our spatial unit of analysis to ensure adequate statistical power, and aggregated median delays across years by locality for each NDC. Connectivity was captured through measures of accessibility (defined as travel times to the "nearest urban centre") for each region in 2015 estimated in Weiss et al [28]. A shapefile of mapped mines was drawn from the 2005 U.S. Geological Survey Mineral resources data system [29]. Finally, a shapefile of Amerindian settlements was obtained from the Land-Mark Global Platform of Indigenous and Community Lands through the GuyNode Spatial Data portal [30].

## Descriptive analysis

We first assessed the stationarity of each region's monthly delay distributions from 2006 to 2019 through an Augmented Dickey-Fuller test. We further examined potential synchronicities between regions' monthly delay distributions, using the mean cross-correlation coefficient between the regional time series for monthly delays from 2006 to 2019.

For each region, we additionally estimated (a) the Pearson correlation between median reporting delays and regional connectivity in 2015, and (b) the Pearson correlation and cross-correlations between monthly total precipitation levels and median delays from 2006–2019. We focused our analysis on case register data from region 4, where many malaria patients come for treatment but there is no local transmission, and from malaria-endemic regions 1, 7, 8, and 9 (see Fig 1A for map).

We used a global Moran's I index [31], to evaluate if delays across NDCs from 2017–2019 (the most recent three years of data) across all ten regions were spatially autocorrelated. This measure estimates whether, and to what extent, observations in the same neighborhood—defined using a range of possible criteria—share "features" which would indicate spatial autocorrelation [31,32]. We further used the Getis-ord statistic, which compares the sum of characteristics in each neighborhood to their overall mean [32,33], to identify local clusters of NDCs with higher reporting delays. To pinpoint the potential correlates of the observed spatial patterns in delays, we computed the local bivariate Moran's I statistic for NDC-level delays and density of Amerindian settlements and for NDC-level delays and density of mines. We chose not to compute the local bivariate Moran's I for NDC-level delays and connectivity, which we expect to exhibit a particularly strong spatial relationship with delays, given that we were restricted to data on connectivity at the scale of regions. For all spatial analyses, we omitted the 2.9% of patients whose localities could not be geocoded. Spatial weights were defined using the queen's contiguity criterion of order = 1. All significance maps for local autocorrelation tests that we provide report FDR-adjusted p-values.

## Nowcasting methods

We used the following two approaches to estimate revised (i.e. eventually reported) monthly malaria case counts in regions 1,4,7,8, and 9. In all of the proposed approaches, our real-time malaria case count estimations were produced via a data assimilation process, in that only information that would have been available at the time of estimation was used to train our models. The first 'out-of-sample' case count estimate for all regions was produced for January of 2007 using historical information available at the time (training set time period) that

consisted of data from the previous 12 months, i.e. within 2006. Subsequent estimates were produced by dynamically training the models below, such that as more information became available the training set consisted of a new set of (12-month long) observations. This approach was chosen as a better way to capture the potential time-evolving nature of reporting delays. For all models, we centered the covariates to ensure standardization of inputs.

## Synchronous data imputation model (DIM)

For each region i, we used an elastic net penalized regression, i.e. an ordinary least squares regression (OLS) subject to a combined L1- and L2- penalty [34] assuming a gaussian error distribution to estimate the revised case count for a given month t. The model was fit to data on the number of cases occurring in months t to t-12 that were known by the end of month t and takes the following form (1), where $\sim \tilde{y}_i(t*)$ denotes the number of cases occurring in each month $t^*$ known by the end of month t. Note that the number of known cases for a given month t increases with the month of separation, as more information accumulates over time. For example, as we would expect to have more information about cases occurring in the prior month than cases occurring in the current month, $\tilde{y}_i(t-1)$ would exceed $\tilde{y}_i(t)$.

$$y_i(t) \sim h_1\tilde{y}_i(t) + h_2\tilde{y}_i(t-1) + h_3\tilde{y}_i(t-2) + \cdots + h_{13}\tilde{y}_i(t-12) + \epsilon \qquad (1)$$

where the $h_i's$ denote the estimated coefficients for each input variable

## Data inputs and method of training and testing the simple data imputation model

A matrix of case counts from each of the twelve months prior to and including time t (the most up to date case count) that were known by the end of time t was used to dynamically train and test the data imputation model in order to predict the number of revised cases for t. Our models were assessed in their ability to accurately predict the revised case counts, only available at time t+m, where m was typically more than a month later, and were compared to the case counts available at time t. We computed a 3-fold cross validation within each twelve month window in order to identify the tuning parameters that resulted in the lowest mean square error and used these parameters for the subsequent (unseen) monthly prediction. In other words, training of our models was conducted using strictly data that would have been available at the time of prediction. Given the size of the training set was 12 observations, we chose a small number of folds, n = 3 for our cross-validation implementation. We experimented with different number of folds (3, 5, or 10) for cross-validation to explore whether this choice would have an impact on the quality of our predictions and found that these additional choices led to qualitatively similar results, so we report the results of the most cost-efficient approach in the manuscript.

## Synchronous network models (NM)

For each region i, an elastic net regression estimating the revised case count for a given month t was calibrated to data on the number of cases occurring in months t to t-12 that were known by the end of month t, the number of cases occurring in months t-1 to t-12 that were known by the end of the month t in all other regions j≠i, and the total precipitation level in region i at month t, a hypothesized cofactor of reporting activity. We also implemented an elastic net regression which additionally takes as inputs the predicted number of cases in month t for all other regions j from the previous data imputation models.

Both network models consider the location where malaria activity is estimated as a node in a network potentially influenced by malaria activity happening in other (potentially neighboring) locations (nodes).

The first network model (NM1) takes the following form (2), where $\theta(t)$ denotes the total precipitation level for region i at time t.

$$y_i(t) \sim \sum_{j \neq i}^{N} h_{1j} \tilde{y}_j(t-1) + h_{2j} \tilde{y}_j(t-2) + \cdots + h_{12j} \tilde{y}_j(t-12)) + \cdots + h_{12N+1} \tilde{y}_i(t)$$
$$+ h_{12N+2} \tilde{y}_i(t-1) + h_{12N+13} \tilde{y}_i(t-12) + h_{12N+14} \theta(t) + \varepsilon \tag{2}$$

The second network model (NM2) takes the following form (3), where $y_j(t)$ denotes the predicted case counts for region j at time t, estimated from the data imputation model for region j (1)

$$y_i(t) \sim \sum_{j \neq i}^{N} h_{1j} \tilde{y}_j(t) + h_{2j} \tilde{y}_j(t-1) + h_{3j} \tilde{y}_j(t-2) + \cdots + h_{13j} \tilde{y}_j(t-12)) + \cdots + h_{13N+1} \tilde{y}_i(t)$$
$$+ h_{13N+2} \tilde{y}_i(t-1) + \cdots + h_{13N+13} \tilde{y}_i(t-12) + h_{13N+14} \theta(t) + \varepsilon \tag{3}$$

Please refer to **S1 Diagram** for a schematic visualizing the distinct and shared inputs of the two network models.

## Data inputs and method of training and testing the two network models

A matrix of case counts from each of the twelve months prior to and including t that were known by the end of t for region i, case counts from each of the twelve months prior to t that were known by the end of t for all other regions j, and total precipitation levels for region i in month t, was used to dynamically train and test NM1 in order to predict the number of revised cases for t for region i.

A matrix of case counts from each of the twelve months prior to and including t that were known by the end of t for region i, case counts from each of the twelve months prior to t that were known by the end of t for all other regions j, DIM-predicted case counts in t for all other regions j, and total precipitation levels for region i in t was used to dynamically train and test NM2 in order to predict the number of revised cases for t for region i.

We replicate the same procedure as in the data imputation model to test and train the two network models.

We generated ensemble model estimates, consistent with the first ARGONet procedure described in Poirier et al[35] by assigning the point estimate for a given month to that of the DIM, NM, and NM2, depending on which model yielded the lowest rRMSE for the prior three months [24,35]. We assigned the mean across models as the point estimate for the first three months of observations.

To provide decision-makers with a prospective uncertainty quantification (error bars) of our estimates, we constructed 95% confidence intervals by considering the error (RMSE) of model predictions for the prior 24 months, starting with January 2009 (two years after the first out-of-sample prediction). Subsequently, following an approach outlined in Poirier et al [35] citing work by Yang et al [36] to capture changing uncertainty at different points in time, we generated the lower and upper limits of the confidence intervals by subtracting or adding the RMSE associated to a moving window of the last 24 errors, respectively, to the upcoming point estimate [35,36]. This confidence interval construction is well-justified as we found that the RMSEs broadly reflected the standard deviation (STD) of the distribution of residuals (predictions—eventually reported or "true" case counts) for each time point (**S7 Fig**) [35,36], and thus, assuming that our

residuals are Gaussian distributed, 95% of residuals should be captured within the mean (the point estimate) plus and minus the STD of residuals.

Lastly, we compared our model to a modeling approach detailed by Bastos et al [17]. We chose to compare to the Bastos et al. model given that it 1) presents a Bayesian "chain ladder" alternative to our models that also considers spatial trends and 2) is implemented for a vector-borne disease in a distinct yet neighboring country context in Latin America (Brazil). For simplicity, we used all priors and hyperparameters as defined in their open access code provided in their manuscript [17].

See **S1 Diagram** for a diagram outlining key steps in this data processing to model implementation process.

All data pre-processing, a-spatial and nowcasting analyses were conducted in R version 4.0.3 [37]. All spatial analyses were conducted in ArcMap version 10.6.1 and GeoDa [38,39].

## Results

### 1. Descriptive analyses

Between 2006 and 2019, malaria cases were entered in the central database a median of 32 days after they were registered at local clinics, with a marked right skew in the distribution of delays, for all regions combined (sd = 52.7). Fig 1B illustrates how these delays occurred in different regions (see **S2 Fig** for full empirical density functions and **S1 Fig** for annual confirmed cases and median delays for each region), and highlights the considerable heterogeneity in the extent of reporting delays observed between them. Due to the pervasiveness of mobile populations, particularly in mining regions, population size estimates and consequently, measures of incidence, are neither meaningful nor tractably quantifiable, so we chose not to report population standardized case counts. Region 1 ranked lowest in the proportion of annually aggregated cases from 2006–2019 reported by the end of the month (median = 0.18, 95% CI = 0.11,0.25) and region 4, where the capital is located, ranked highest, reporting nearly all of its cases by the end of the month (median = 0.93, 95% CI = 0.87,0.97). Neighboring regions 7, 8 and 9 additionally captured a low fraction of cases by the end of the month (median = 0.49, 95% CI = 0.30,0.60, median = 0.24, 95% CI = 0.12,0.51; median = 0.30, 95% CI = 0.22,0.41, respectively).

Monthly total precipitation was not correlated with monthly median delays between 2006 and 2019 for any of the regions (p-value = 0.90, p-value = 0.80, p-value = 0.41, p-value = 0.62, p-value = 0.84, for regions 1,4,7,8 and 9, respectively). While we expect that the differences between region 4 and the other regions in reporting delays are likely due to the remoteness of regions 1, 7, 8, and 9, the estimated association between region-specific connectivity and median delays in 2015 was modest and non-significant (r = -0.13, p-value = 0.71). Total precipitation and monthly delays for region 4 reported a significantly (at a level = 0.05) negative cross-correlation of -0.18 at a lag of -1 months, indicating that increased precipitation precedes reduced delays by one month in region 4. For all other regions, we found no significant cross-correlations within a meaningful range in lags.

We observed significant spatial autocorrelation in median reporting delays by NDC for the most recent years of data (Global Moran's I = 0.496,p = 0.001). NDCs characterized by long delays were found in close proximity to NDCs with similarly high delays and vice versa (**S3 and S4** Figs). Significant clusters of NDCs marked by higher delays were most concentrated in regions 1 and 7, while clusters of NDCs marked by lower delays were most concentrated in regions 4 and 6 (**S3** and **S4** Figs). Delays were significantly positively spatially correlated with density of mines by NDC, with the bivariate analysis highlighting areas with NDCs characterized by elevated delays surrounded by a greater density of mines (largely concentrated in

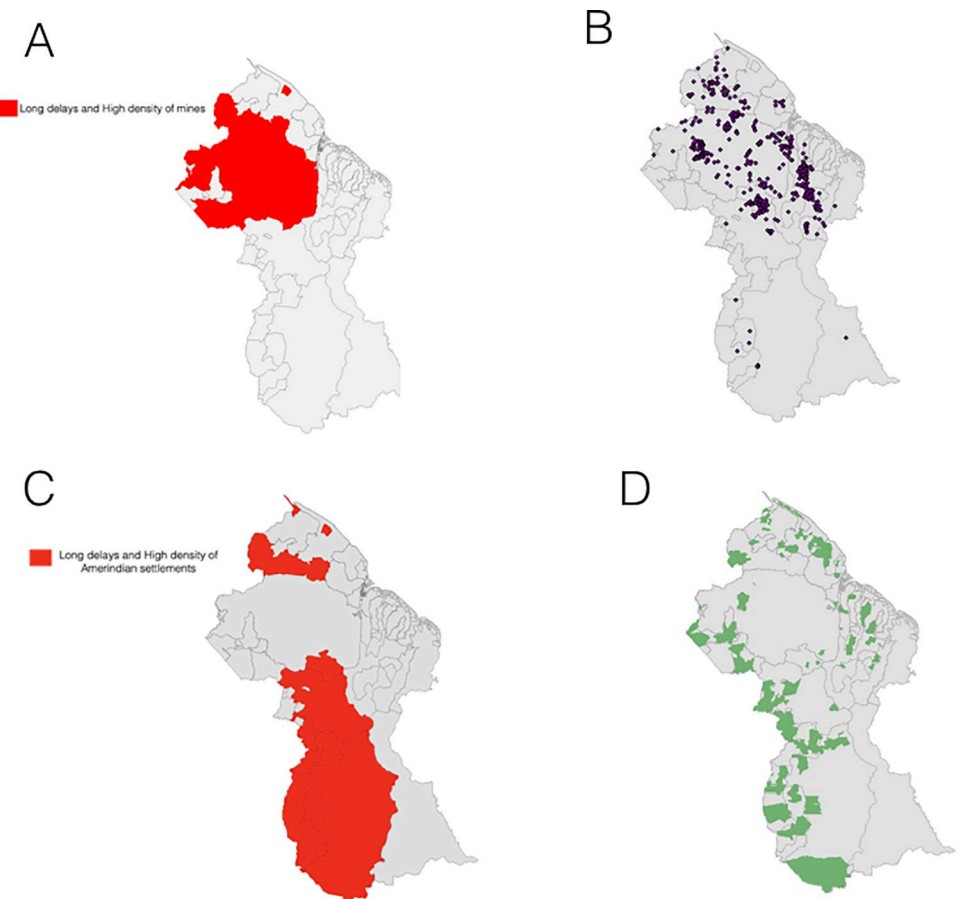

**Fig 2. A. Local bivariate Moran's I clustering map of aggregated delays and density of mines, by NDC; B. Map of mining sites; C. Local bivariate Moran's I clustering map of aggregated delays and density of Amerindian settlements, by NDC; D. Map of Amerindian areas. For visual purposes, we show only high-high clusters of areas reporting at higher delays with a greater density of mines or Amerindian settlements. Full bivariate cluster and significance maps can be found in S5 Fig.** *Base map sourced from DIVA-GIS.*

regions 1 and 7) (**Figs 2A, 2B, S5A and S5B**) (bivariate Moran's I = 0.590, p = 0.001). We also found evidence for a significant positive bivariate spatial association between reporting delays and density of Amerindian settlements, revealing areas with NDCs characterized by higher delays surrounded by a greater density of Amerindian settlements, mainly concentrated in regions 1, 8 and 9 (**Figs 2C, 2D, S5C and S5D**) (bivariate Moran's I = 0.587, p = 0.001). Both clustering maps signal NDCs within region 1 marked by higher reporting delays and a relatively increased presence of Amerindian settlements and mines. Note that NDCs showing the inverse of this relationship, i.e. lower delays surrounded by a higher density of Amerindian settlements or mines, were located in regions 6 and 10, where malaria is not endemic (**S5A–S5D Fig**).

We found weak evidence for overall synchronicity in delay distributions across regions, with an estimated mean cross-correlation between the regional monthly delay distributions equal to 0.24 (95% CI = 0.20,0.27) and to 0.25 (95% CI = 0.18,0.41) when excluding region 9, due to low case numbers. Overall, these minimally correlated delay distributions may reflect varying seasonality in transmission between regions. Finally, there was significant evidence for stationary of the time series of monthly delay distributions for all regions (from 2006–2019)

(p-value = 0.01 for all six regions). These findings support the use of data imputation models, which rely on previous trends in delays in a given region to inform ongoing predictions.

## 2. Nowcasting results

**A. Data Imputation Models (DIM).** The model based on region 4, where the capital Georgetown is located, yielded the best estimate of monthly cases from 2007 to 2019 (rRMSE = 0.0781). DIM model predictions coincided with the two highest peaks in true cases and generally reflected trends throughout time, despite substantially underestimating a moderate peak in late 2012 and generally overshooting revised cases in late 2013 (**Fig 3B**). Given that the number of cases known by the end of each month in region 4 closely approximated the number of revised cases, this approach may be of limited practical utility. Due to the high accuracy of region 4 DIM, we chose not to implement network models for this region.

In contrast, the DIM models for regions 1, 7 and 8 significantly improved on revised case counts for each of these regions. Of these, the DIM model for region 8 reported the optimal performance (rRMSE = 0.151), exhibiting a slight lag in estimating revised cases, sharply undershooting cases in late 2011 and early 2013, while eclipsing cases in late 2013, but closely recapitulating trends from 2014 onwards (**Fig 3D**). The DIM models for regions 1 and 7 reported weaker model performance (rRMSE = 0.270 and 0.174, respectively), partially driven

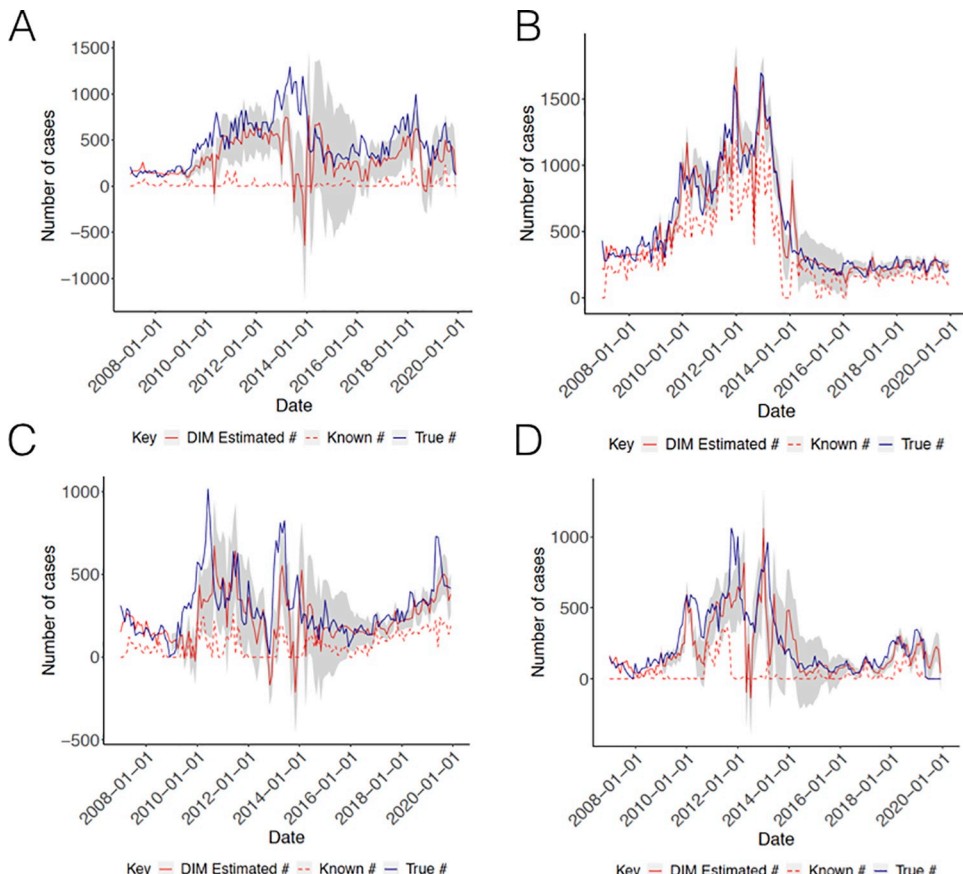

**Fig 3. Results from Data Imputation Models.** (A) Regions 1 (B) Region 4 (C) Region 7 and (D) Region 8. Solid red lines indicate the number of cases estimated from the data imputation model, dashed red lines indicate the number of cases known by the end of the month and solid blue lines indicate the true number of eventually reported cases. Grey bands capture the moving 95% confidence intervals for each model.

by the marked underestimation of revised cases in early 2011, late 2013, and late 2019, in the case of region 7, and the consistent underestimation of revised cases over the course of the study period, in the case of region 1 (**Fig 3A and 3C**). However, region 1 predictions broadly approximated converged case trends, although still of an attenuated magnitude, from 2017 onwards, and model predictions for region 7 still generally reflected the most recent upward trend in cases from 2016, with the exception of late 2019, improving on month-end reporting substantially (**Fig 3A and 3C**). Finally, the DIM model for region 9 failed to converge (identify a stable set of parameter values) due to data sparsity, largely driven by the exclusion of cases with implausible dates of documentation, as defined previously. Of note, the data imputation model estimates for all regions, particularly for region 1 and 7, were marked by wide confidence intervals from 2014–2016, a consequence of the poor model performance in the years just preceding. However, it is important to highlight that we observe consistently sharper confidence intervals, and thus an increased level of certainty in our estimates, for cases occurring in the more recent past, i.e. from 2016 onwards. Finally, for all regions, a slight majority of monthly converged case counts fall within our estimated confidence intervals, testifying the effectiveness of our instituted method for prospectively estimating confidence intervals.

**B. Network Models (NMs).**   Both network models exhibited improvements for all regions (**Fig 4A–4D**). The NMs for region 1 resulted in predictions that more closely tracked revised case counts over time (**Fig 4A and 4B**), showing reduced error rates for both models (rRMSE = 0.2475 [NM1] and 0.2490 [NM2]), compared to the data imputation models. However, both models demonstrated continued underestimation of case counts from 2011 to 2014. NM1 predictions for region 1 reported only marginally greater accuracy than the corresponding predictions from NM2. The NMs for region 7 revealed perceptible gains in accuracy, more closely reflecting true dynamics in case counts and importantly, for both models, attenuating the appreciable drop in estimated cases in late 2012 and correcting estimates to better mimic case trajectories from 2016 (**Fig 4C and 4D**), with NM1 resulting in the greatest improvement (rRMSE = 0.1706 [NM1] and rRMSE = 0.1718 [NM2]). Finally, NM1 and NM2 improved over the DIM model for region 8 (rRMSE = 0.1387 and 0.1359, respectively). Model predictions from late 2013 onwards better mirrored the magnitude and timing of current trends (**Fig 4E and 4F**). Unlike regions 1 and 7, model predictions were most improved for NM2.

The data imputation and network models significantly improved over low known case counts for regions 1, 7, and 8, most strikingly for regions 1 and 8, with error rates reduced by a factor of 1.8, 1.6, and 2.1, respectively, for the regions' best performing models (**S1 Table**). In all cases, when relying on predictions from the best performing models, the median fraction of cases captured within a month for all regions from 2007 to 2019 now ranges from 0.63 to 0.95.

**C. Ensemble results.**   For all regions, excluding region 8, the two network models were outperformed by their corresponding ensemble models, reporting slightly increased rRMSEs compared to their network model counterparts (rRMSE = 0.2379 [Region 1 ensemble], 0.1610 [Region 7 ensemble], and 0.1399 [Region 8 ensemble]; **S1 Table** and **S6A–S6C Fig**). The relatively increased accuracy associated with the network models for region 8 (compared to their ensemble counterpart) was modest, such that the increased interpretability or operational ease of reporting a singular point estimate that takes into account the predictions across the three models likely may offset any costs in model performance. Further, while each of the network models alone proved to be more accurate than the ensemble over our period of interest, the NMs may not consistently perform better during subsequent periods, rendering decision-making processes difficult, due to the constant updating and re-identification of the optimal model. By contrast, the ensemble model is capable of dynamically selecting the best performing model based on recent predictions and then arriving at a single point estimate on its own, precluding the need for decision-makers to gauge the relative accuracy of each of the models

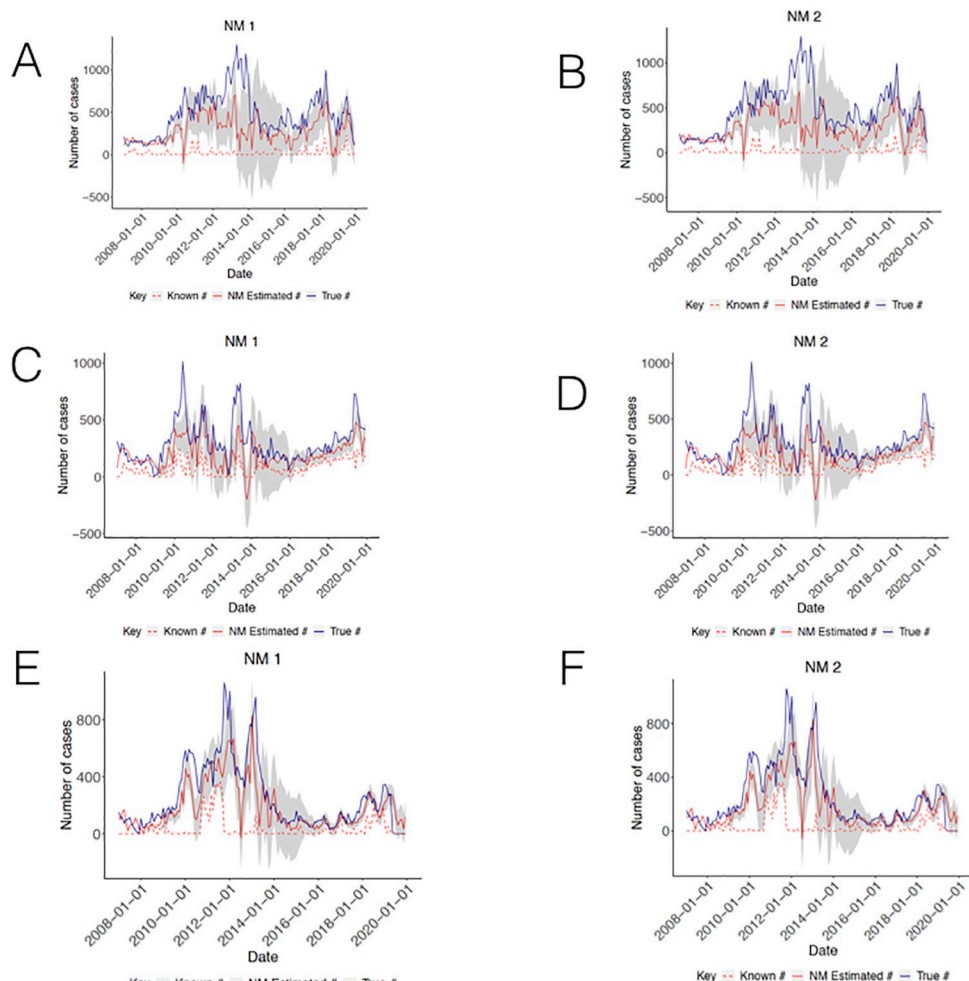

**Fig 4. Results from Network models.** (A) Region 1 NM1 (B) Region 1 NM2 (C) Region 7 NM1 (D) Region 7 NM2 (E) Region 8 NM1 (F) Region 8 NM2.

after the fact. It is however important to clearly detail the process by which the ensemble model ranks the different models and generate point estimates at each time step, to ensure that the process of arriving at ensemble estimates is transparent to all health officials. It is also worth underscoring that the ensemble selection process occurs almost instantaneously, mitigating any potential concerns about increased model run times for generating ensemble nowcasts. Despite these relative advantages of the ensemble models, the number of months over which error rates are compared is, although informed by related studies [24], somewhat an arbitrary choice and comparisons over the past three months may not necessarily imply an accurate estimate in the present month, notably when reporting trends are so erratic.

**D. Comparison to an existing approach.**   In comparison to Bastos et al. model [17], the DIM models for regions 1,4, and 7, i.e. the models with the highest error rates for each region, exhibited greater accuracy, reporting RMSEs for monthly aggregated median estimated case counts of 0.2958, 0.1336, and 0.3405, respectively. In contrast, the Bastos et al. model estimates for region 8 reported a lower rRMSE [rRMSE = 0.0782] than our corresponding DIMs and NMs, indicating marked improvements in performance of the Bastos et al. alternative. The Bastos et al. model was also able to generate nowcasts for region 9, indicating its potential use

in regions with exceedingly sparse data. Additionally, their modeling framework provides nowcasted estimates at the weekly level, which may be of use during times when more rapid surveillance and response activity is required; of course, this does come at the costs of increased error rates given the little information on eventually reported cases known within a week. Run times for the Bastos et al. models for each region were noticeably longer than our DIMs and NMs, but not excessively so, and is thus not necessarily a key deciding factor against its implementation. However, one key challenge of the effective use of this framework is the communication of its underlying tenets to health officials and surveillance officers not familiar with Bayesian approaches, particularly which priors to specify and how approximate Bayesian computation is involved to arrive at final outputs.

## Discussion

Our findings highlight the extent to which reporting lags pose a challenge to malaria surveillance in Guyana, with appreciable heterogeneities in the overall magnitude and time trends of delays across areas, and the potential social factors linked to these observed delay patterns. Such trends are likely to be recapitulated in similar settings with fragmented surveillance systems and infrastructural challenges. While improvements to reporting systems, including digitization, infrastructure, and enhanced data management, are critical for malaria control in the medium- and long-term, many national programs will not have the resources to make significant changes in the near future. The nowcasting approach we describe here therefore represents a simple analytical method leveraging data that is often already being collected in order to significantly improve estimation of current malaria cases, at almost no increased investment. Our approach provides an easily accessible tool, which necessitates no additional data requirements and minimal technical training, and can be readily implemented in resource-constrained surveillance contexts.

We found convincing evidence for spatial clustering in delay patterns at the level of Neighborhood Democratic Councils, while synchronicities between reporting delays at the level of regions was more moderate. The minimal overlap of regional delay patterns may partially explain the modest improvements when incorporating time trends and predicted case counts in other regions to estimate region-specific case counts as done by the network models. Finally, we observed significant spatial associations between the presence of Amerindian settlements and mines and reporting delays, primarily occurring in regions 1, 7, 8 and 9, with the two overlapping in region 1. This finding suggests that issues in timeliness of reporting are most acutely distributed among socially remote and resource-limited communities.

In contrast, associations between precipitation and delays (from 2006–2019) were marginal or non-intuitive (i.e. increased rainfall associated with lower delays). We note that this may be partially attributed to the fact that increased rains affecting driving conditions may undermine reporting timeliness at both a finer spatial scale than regions, or even neighborhood democratic councils, (i.e. with particular roads and towns threatened due to locally-specific infrastructural challenges) and at a finer temporal scale than months. Thus, these patterns are obscured in region- and monthly-level associations. Furthermore, the source of interpolated precipitation data may not fully represent real-world behavior, as "regions with sparse support [are affected by] patchy coverage", as is the case in several areas within Guyana [26]. Finally, the association between connectivity and delays in 2015 was minimal and non-significant. This finding can be explained by the significant variation in the level of connectivity—both in terms of road access and other forms of travel—observed within regions, which is again diluted in region-level associations.

The data imputation models reported relatively low error rates, suggesting that more simple models for each region that simply learn from their own autoregressive trends may be practical

and straightforward to implement. While there was considerable variation in DIM model performance across regions, it is promising that the model achieved improvements on underestimated cases in regions 1 and 8, the regions with the lowest fraction of cases reported within a month, and thus have the greatest need for improved real-time surveillance. Furthermore, while the network models for all regions vastly outperformed the corresponding data imputation models over the entirety of the study period, ensemble models which dynamically processes predictions from all models to pinpoint the ideal model may be an easier alternative to report. However, issues related to selecting the error processing period to consider and effectively communicating the mechanics of the ensemble to a broader audience, as previously discussed, should be paid heed. Possible extensions to the network models to markedly improve their predictive power include accounting for other covariates that may better explain trends in reporting rates, such as access to waterways, another important source of travel. Additionally, the NMs may be best implemented using data on the previous distribution of known case counts by NDC (rather than by region), or at a more local spatial unit for which stronger relationships in reporting delays across locations are observed. Finally, to address the presence of negative case predictions, we explored log transforming the target observations, but this methodological approach did not systematically improve the performance of our models.

There are a number of additional constraints of our modeling approach that should be addressed in future applications of our work and extensions to other settings. First, our models are not well-suited for uncovering case counts for newly emerging diseases, as they require sufficient historical data to feasibly learn from reporting trends. Additionally, our models do not allow flexible and non-linear relationships between the number of revised cases in a given region in month t and each the relevant predictors, such as potential interactions between the known case counts in month t-2 and month t-3, or between known case counts in month t-4 and precipitation (or other climate-related and other relevant covariates) in month t. In future work, non-parametric methods like regression trees can instead be implemented to avoid these potentially erroneous model assumptions, fitting ensemble methods like random forest regression trees to data on prior known case counts in each region and total precipitation levels to identify those which contribute most significantly to accurate estimates. Such methods may be particularly useful when including multiple additional variables, such as climatological factors or other informative inputs beyond case reporting data. However, it is worth noting that ensemble tree-based approaches may come with the added cost of reduced accessibility, in contrast to more model-based approaches like elastic net regression. Furthermore, such methods require a number of decisions on hyperparameters, such as the number of trees used, tree depth, fraction of input variables considered, among others, in contrast to the sole choice of the lambda and alpha parameters for our elastic net models.

It is important to underscore that our approach can only be applied when there is sufficient data for model training and testing, as all models for region 9 failed to run due to sparse data (total number of cases and thus, cases known at each prior time point), limiting its application to areas with a sufficient number of malaria cases. However, despite these challenges, we believe that data imputation and network model approaches, which learn from time and spatial patterns in reporting, can greatly improve the current state of surveillance efforts, particularly for areas where data quality is poorest and remoteness lead to significant reporting delays. We note that due to the moving training window approach, our model is intrinsically able to capture existing reporting trends each year, i.e. scale of underreporting, over time. Our estimates are currently being integrated in the National Malaria Programme in Guyana, which has prioritized surveillance as a key intervention to help in the achievement of malaria elimination. Their use has afforded the opportunity for policymakers to use close-to-accurate estimates to drive planning, resource allocation and implementation of malaria elimination

activities. The potential for impact is far-reaching and serves as a suitable tool for use in other countries within the Guyana Shield as well as the Region of the Americas.

## Supporting information

**S1 Fig. Annually aggregated cases recorded at health facilities by region, from 2006–2019. (A) For all patients (B) Excluding patients with implausible recorded date ranges, i.e. were known to be cases at local hospitals after they were known to be cases at the central office in Georgetown).** Plot labels indicate the median of delays for all cases from 2006–2019 for each region. Note the relatively low case counts in region 9 from 2016 onwards, particularly after excluding cases with implausible date ranges, lending to small cell counts when parsing the number of cases occurring in a given month known by the end of each subsequent month and the consequent failure for the region 9 models to converge.
(TIFF)

**S2 Fig. Distribution of delays from 2006 to 2019 for regions 1, 4, 7, 8 and 9.**
(TIFF)

**S3 Fig. Map of median delays reported by region (darker colors indicate higher median delays).** Base map sourced from DIVA-GIS.
(TIFF)

**S4 Fig. Local G\* clustering maps of median delays. (A) Map of G\* statistic (B) FDR-adjusted significance map.** *Base map sourced from DIVA-GIS.*
(TIFF)

**S5 Fig. Local bivariate Moran's I clustering maps. (A) Aggregated median delays and density of mines (B) FDR-adjusted significance map. (C) Aggregated median delays and density of Amerindian settlements (D) FDR-adjusted significance map. Source for base map:** https://www.diva-gis.org/datadown.
(TIFF)

**S6 Fig. Results from all models: DIM (red, solid), NM1 (magenta), NM2 (cyan), and the ensemble model (green). (A) Region 1 (B) Region 7 (C) Region 8.** As before, dashed red lines indicate the number of cases known by the end of the month and solid blue lines indicate the true number of eventually reported cases. Grey bands capture the moving 95% confidence intervals for each model.
(TIFF)

**S7 Fig. Histogram of monthly residuals (model predicted- eventually reported cases) from January 2007—December 2019 and overlaid Gaussian density curve for (A) Region 1 DIM (B) Region 7 DIM (C) Region 8 DIM (D) Region 1 NM1 (E) Region 7 NM1, (F) Region 8 NM1 (G) Region 1 NM2 (H) Region 7 NM2 (I) Region 8 NM2. (J) Region 1 Ensemble (K) Region 7 Ensemble (L) Region 8 Ensemble.** All panels report the rmse of the corresponding model predictions and standard deviation of the corresponding residuals. Histogram + Gaussian density curve for region 4 DIM (rmse = 122.38 and sd = 122.00). (follows the format provided in Poirier et al [35].
(TIFF)

**S1 Table. Table of relative root mean squared errors (rRMSE) generated from data imputation models and network models for each region.** Note: for region 4, rRMSE only reported for the data imputation model, given that no network models were run for this region). The final row reports the best performing model\* for each region, excluding the ensemble, which

corresponds to the model resulting in the lowest rRMSE.
(DOCX)

**S1 Diagram. Flowchart detailing each of the steps in the model implementation process (for each region), from data processing, to preparation of the input data frame, to the models run.**
(TIFF)

**S2 Diagram. Network model schematics. (a)** Diagram detailing each of the components of the first network model, i.e. monthly known case counts up to t-1 for all regions j≠i and for known case counts up to t for region i (scroll, shaded in light green) and monthly precipitation levels for all regions j in month t (cloud with rain, shaded in light blue) **(b)** Diagram detailing each of the components of the second network model, i.e. monthly known case counts up to t-1 for all regions j≠i and for known case counts up to t for region i (scroll, shaded in light green), DIM predicted case counts for month t for all regions j≠i (cylinder, shaded in dark green), and monthly precipitation levels for all regions j in month t (cloud with rain, shaded in light blue). For both a and b, we use region 7 as the example region of interest for which we are producing the network model estimates.
(TIFF)

## Acknowledgments

This work is dedicated to Sharon Jones-Weekes, who has devoted more than twenty years of service and research to the Guyana Vector Control Services surveillance program.

We thank Ayesha Mahmud and Nishant Kishore for their helpful suggestions and support and Marcia Castro for providing guidance during early explorations of some of our spatial analyses. We are also indebted to the entire VCS team in the central office and Bartica District Hospital for all their work in facilitating this data collection effort and for detailing and contextualizing the process of reporting.

## Author Contributions

**Conceptualization:** Tigist F. Menkir, Horace Cox, Mauricio Santillana, Caroline O. Buckee.

**Data curation:** Tigist F. Menkir, Horace Cox, Melanie Saul, Sharon Jones-Weekes, Collette Clementson, Pablo M. de Salazar.

**Formal analysis:** Tigist F. Menkir, Canelle Poirier, Mauricio Santillana, Caroline O. Buckee.

**Funding acquisition:** Caroline O. Buckee.

**Investigation:** Tigist F. Menkir, Horace Cox, Melanie Saul, Sharon Jones-Weekes, Collette Clementson, Pablo M. de Salazar, Mauricio Santillana, Caroline O. Buckee.

**Methodology:** Tigist F. Menkir, Mauricio Santillana, Caroline O. Buckee.

**Project administration:** Horace Cox, Collette Clementson, Mauricio Santillana, Caroline O. Buckee.

**Resources:** Horace Cox, Melanie Saul, Sharon Jones-Weekes, Collette Clementson.

**Software:** Tigist F. Menkir, Canelle Poirier.

**Supervision:** Horace Cox, Mauricio Santillana, Caroline O. Buckee.

**Validation:** Tigist F. Menkir, Canelle Poirier, Mauricio Santillana.

**Visualization:** Tigist F. Menkir.

**Writing – original draft:** Tigist F. Menkir.

**Writing – review & editing:** Tigist F. Menkir, Horace Cox, Canelle Poirier, Melanie Saul, Collette Clementson, Pablo M. de Salazar, Mauricio Santillana, Caroline O. Buckee.

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
