## [Decision Letter · Decision Letter 0]

27 May 2021

Dear Dr. Buckee,

Thank you very much for submitting your manuscript "A nowcasting framework for correcting for reporting delays in malaria surveillance" for consideration at PLOS Computational Biology.

As with all papers reviewed by the journal, your manuscript was reviewed by members of the editorial board and by several independent reviewers. In light of the reviews (below this email), we would like to invite the resubmission of a significantly-revised version that takes into account the reviewers' comments.

All reviewers agree that the manuscript introduces interesting and

important ideas that could advance the methodology available

to correct for delays in malaria case reporting and thus improve

decision making regarding control strategies. Reviewers 1 and 4

identify important issues requiring the attention of the

authors. Among them I highlight the importance to address:

i) the potentially misleading claims about alternative approaches

ii) the need to report measures of uncertainty regarding the newly

introduced methodological approaches

iii) the need to provide performance comparison among approaches

iv) literature review

v) make code available

vi) provide details regarding model description

In addition to the above, remaining issues brought up by the reviewers

should also receive careful attention by the authors.

We cannot make any decision about publication until we have seen the revised manuscript and your response to the reviewers' comments. Your revised manuscript is also likely to be sent to reviewers for further evaluation.

Sincerely,

Claudio José Struchiner, M.D., Sc.D.

Associate Editor

PLOS Computational Biology

Thomas Leitner

Deputy Editor

PLOS Computational Biology

All reviewers agree that the manuscript introduces interesting and

important ideas that could advance the methodology available

to correct for delays in malaria case reporting and thus improve

decision making regarding control strategies. Reviewers 1 and 4

identify important issues requiring the attention of the

authors. Among them I highlight the importance to address:

i) the potentially misleading claims about alternative approaches

ii) the need to report measures of uncertainty regarding the newly

introduced methodological approaches

iii) the need to provide performance comparison among approaches

iv) literature review

v) make code available

vi) provide details regarding model description

In addition to the above, remaining issues brought up by the reviewers

should also receive careful attention by the authors.

Reviewer's Responses to Questions

**Comments to the Authors:**

Reviewer #1: Review uploaded as an attachment.

Reviewer #2: general comments:

I believe the presentation of the methodology could greatly benefit from the adition of a figure containing some kind of flow diagram of the entire analysis from the Imputation through to the Nowcasting.

A basic aspect of malaria epidemiology, is that it may persist on the same patient for a long time with many relapse episodes. Thus when a temporal cluster of cases is observed, it is not necessarily indicative of an outbreak of new infections. Besides, clusters of case reports may also be influenced by observational biases, for example when health professionals visit a remote village and test the whole community in a short interval of time. The authors must explain how they hope to describe incidence patterns given these confounding factors. Some reported cases may also be a relapse of the same patients.

Regarding the Nowcasting models, I think that calling the first regression model an "imputation" is not entirely accurate because the model actually estimates cases which have occurred but not yet been reported at time t. Imputation is when you replace missing data with something else. Even though these revised case counts are used in the subsequent models, this alone is different from a standard data imputation scheme.

The authors then describe two types of "network" models that also include the previous 12 months of data from all other regions to predict for a given region. Agin here I think that calling a network model is misleading since no particular influence network between the regions is being considered, instead all the regions are included. Moreover, from the text alone, the distinction between the first network model and the second is not easy to grasp, I think the authors should either improve the textual description or add a figure to help tease apart the two models' structures.

In the first paragraph of the Nowcasting section, the authors say that "...The first out-of-sample case count estimate for all regions was produced for January of

2007 using historical information available at the time (training set time period) that consisted of data from the previous 12 months (within 2006)..." And they also say that "Subsequent estimates were produced by dynamically training the models... as more information became available...". If they continued to train the model on the new data, then none of the estimates produced by the model were trully "out of sample". This statement must be better justified or corrected. A more traditional out-ofsample validation scheme should be presented where the validation set would be separated beforehand and used only to for predicting purposes, once the training is done. There's nothing wrong with a rolling window within the training set.

Confidence intervals for the the revised case counts should be added to figures 3 and 4.

Reviewer #3: The authors have analyzed 13 years of malaria surveillance data from Guyana. They developed methods to "nowcast" cases, meaning that they used the historical data as it would have been available to the program, and they developed methods to accurately predict the actual number of cases that would eventually be reported. The idea is not particularly new, but the study was well-done. The paper was well-written. The idea is important and interesting. I recommend accepting the article, though I would like to make a few suggestions to the authors to be considered (or rejected) as they wish.

First, I think it would be good to have a simple table summarizing some very basic things. How many people lived in each district? How many malaria cases were reported from each district in each year? Thus what was the incidence of confirmed malaria cases, measured as incidence per 1,000 population, per year? What was the median delay for each district?

Second, it's dissatisfying that the models failed to converge for district 9. It looks like it might be the district where the method would be most useful. Is there anything you can do? The supplementary figure S1 makes it look like there was a pretty reasonable pattern in the delay for Region 9, so I'm scratching my head about why. If the problem is that there were too few cases, then that line in the table I suggested would make it clear if it was a low number problem.

Third, I note that you have three methods (for districts 1, 7, and 8) that work well. Sometimes having three good methods is worse than having just one, if only because it creates confusion about which one to use. I think it would increase the cool factor of the paper and also probably help programs know what to do if you created an ensemble prediction combining the three methods.

Fourth, if the dashed red lines and the solid blue lines are same for every district in Figures 3 and 4 (for districts 1,7,and 8), why not consolidate? It would be easier to see the differences in the predictions made by various methods (give each method a different color).

I hope these will be considered suggestions by both editors and authors and not cause the authors stress or grief. If what I'm suggesting is out of line, please explain why it's not possible.

Reviewer #4: "A nowcasting framework for correcting for reporting delays in malaria surveillance" by Menkir et al. is a manuscript that proposes a method to anticipate malaria cases in Guiana based on historical data. Methods for delay correction have been applied to HIV, influenza, dengue fever, and more recently in COVID-19. As far as I am concern it is the first time it has been implemented in malaria, where notification delay is an important issue to be dealt with in an endemic region. Malaria endemic countries in general lack good infrastructure in countryside regions leading to large delays, Guiana faces the same issue. The manuscript aims to tackle an important issue in Guiana and the authors deliver a solution which I think it is indeed simple and useful however it lacks a better literature review on nowcasting methods and also a comparison between the proposed methods and the existing ones (or a justification for not doing this comparison), and a better description of the proposed models is needed. On the good side, their proposed model has a good performance both visually and using the rRMSE.

Some comments:

On page 2 the authors said "One limitation of the Bayesian methods is that they do not focus on providing more interpretable measures to guide actionable surveillance efforts, such as direct point estimates for predicted case counts." The Bayesian methods described are all based on the chain-ladder method, a relatively well-known tool in actuarial sciences used to calculate incurred but not reported (IBNR) loss estimates (Renshaw, 1989). The models mentioned in the manuscript (Rotejanaprasert t al., 2020; McGough et al., 2020; Bastos et al., 2019). There are both Bayesian and non-Bayesian approaches and I failed to see the described limitation since in all recent recent approaches one can have more than a point estimate of case counts, they all can provide a predictive distribution of the case counts.

The literature review of nowcasting models in infectious diseases should also include Salmon et al. (2015) and Stoner and Economou (2020), they both provide extensions of the chain-ladder model and apply their proposed methods into infectious diseases data. More recently there are these extensions trying to model COVID-19 deaths (Hawryluk et al., 2020; Seaman et al.; 2020). I am not suggesting an exhausting literature review on methods, but there is some literature on the matter and I expected some comparison to existing methods, or perhaps a better description saying why the existing methods are not suitable in these context,

Data description: Could the authors describe the data? How is the delay calculated? On page 5 there is odd notation `~\\bar{y}_i(t*)` that "denotes the number of cases occurring in each month t* known by the end of month t". Should that be \\widetilde{y}_i(t*)? But where is t in this notation? How can \\widetilde{y}_i(t*) at time t-1 be different than \\widetilde{y}_i(t*) at time t? I might be confused with the notation here. It would be important to make it clearer.

About the models: DIM is a baseline model using as covariates the number of cases for the past twelve months observed at time t, in addition to that there is an assumption that these coefficients do not vary in space, since h is not indexed by region. NM-1, the {h} coefficients are not indexed by region, is that correct? I believe they vary by region, i.e. y_i(t) \\sim \\sum_{j \\neq i} \\sum_k h_{(j-1)*12 + k} \\widetilde{y}_{j}(t-k). This is also valid for model NM-2, which extends model NM-1 by adding N-1 covariates, the predict cases of regions j different than i using model DIM. Is that correct? The notation in equation (1-3) is not clear, another common notation in a modelling approach is the use of tilde y ~ a*x + b*z, but is there probability distribution behind the elastic net penalized model? By using the OLS can I assume a Gaussian distribution is assumed in the background?

Uncertainty: The uncertainty of the predictions is ignored. It should have a justification for ignoring it. The authors consider only a predictive point estimate. I assume that their proposed methods allow to estimate intervals for the predictions (nowcasts). Hence I would expect a uncertainty interval for the solid red lines in Figures 3 and 4. For models DIM and NM-1 I assume the intervals are straightforward to obtain, for NM-2 is not so direct because there is some uncertainty on the predictive value used as a covariate that should be propagated.

Code and data sharing: Would R code and data be available? Even if the authors opt to not compare their methods with other nowcasting methods it would be very interesting to evaluate the performance of the elastic net penalized regression for nowcasting.

Minor comments:

Equation 1: The subscripts i on the left hand side and l (?) on the right hand side of the equation need to be describe.

Page 6: There is no such version R version 1.2.1335, the version 1.2.2 was released in 2001. I assume 1.2.1335 refers to RStudio version, in order to find out the R version type "version" on R console.

Refs:

Hawryluk, Iwona, et al. "Gaussian Process Nowcasting: Application to COVID-19 Mortality Reporting." arXiv preprint arXiv:2102.11249 (2021).

Renshaw, A.E. "Chain ladder and interactive modelling (claims reserving and GLIM)." Journal of the Institute of Actuaries (1886-1994) 116.3 (1989): 559-587.

Salmon, M., Schumacher, D., Stark, K. and Höhle, M. (2015) Bayesian outbreak detection in the presence of reporting delays. Biometrical Journal, 57, 1051-1067.

Seaman, Shaun, et al. "Nowcasting CoVID-19 Deaths in England by Age and Region." medRxiv (2020).

Stoner O, Economou T. (2020) Multivariate hierarchical frameworks for modeling delayed reporting in count data, Biometrics, volume 76, no. 3, pages 789-798, DOI:10.1111/biom.13188.

**Have the authors made all data and (if applicable) computational code underlying the findings in their manuscript fully available?**

Reviewer #1: **No: **The authors did not appear to provide code, which will hinder the study and use of their approach by other researchers and practitioners.

Reviewer #2: **No: **

Reviewer #3: Yes

Reviewer #4: **No: **The authors have mentioned the some restrictions will apply.

PLOS authors have the option to publish the peer review history of their article (what does this mean?). If published, this will include your full peer review and any attached files.

Reviewer #1: No

Reviewer #2: No

Reviewer #3: No

Reviewer #4: **Yes: **Leonardo Bastos (Oswaldo Cruz Foundation)
---

## [Decision Letter · Decision Letter 1]

23 Sep 2021

Dear Dr. Buckee,

Thank you very much for submitting your manuscript "A nowcasting framework for correcting for reporting delays in malaria surveillance" for consideration at PLOS Computational Biology. As with all papers reviewed by the journal, your manuscript was reviewed by members of the editorial board and by several independent reviewers. The reviewers appreciated the attention to an important topic. Based on the reviews, we are likely to accept this manuscript for publication, providing that you modify the manuscript according to the review recommendations.

Reviewer 4 indicates one minor additional point that might improve the paper. The authors might find it worth considering.

Sincerely,

Claudio José Struchiner, M.D., Sc.D.

Associate Editor

PLOS Computational Biology

Thomas Leitner

Deputy Editor

PLOS Computational Biology

[LINK]

Reviewer 4 indicates one minor additional point that might improve the paper. The authors might find it worth considering.

Reviewer's Responses to Questions

**Comments to the Authors:**

Reviewer #1: I am satisfied that the reviewers have addressed all of my comments.

Reviewer #3: I have no further comments or suggestions. I think the authors responded to all the suggestions in a way that strengthened the paper.

Reviewer #4: The authors have answered all of my questions, and I am happy with the manuscript. It is well-written and tackles a relevant public health problem not only for Guiana, but for all countries where malaria is endemic and there is a case notification system.

I would like to make a suggestion though. The estimates for both models data imputation model (DIM) and networks models (NM1 and NM2) return negative estimates for the number of cases (Figures 3 and 4). So I wonder if use a log-transformation in the modelling approach and then transform back to the original scale for the estimates would improve the results. It would certainly avoid the negative estimates, but I am not sure if it would somehow mess up with the uncertainty. Since it is a cheap model to run, it may be worth testing.

**Have the authors made all data and (if applicable) computational code underlying the findings in their manuscript fully available?**

Reviewer #1: None

Reviewer #3: Yes

Reviewer #4: Yes

PLOS authors have the option to publish the peer review history of their article (what does this mean?). If published, this will include your full peer review and any attached files.

Reviewer #1: No

Reviewer #3: No

Reviewer #4: No

Figure Files:

Data Requirements:

Reproducibility:

References:

---

## [Editor Report · Decision Letter 2]

18 Oct 2021

Dear Dr. Buckee,

We are pleased to inform you that your manuscript 'A nowcasting framework for correcting for reporting delays in malaria surveillance' has been provisionally accepted for publication in PLOS Computational Biology.

Best regards,

Claudio José Struchiner, M.D., Sc.D.

Associate Editor

PLOS Computational Biology

Thomas Leitner

Deputy Editor

PLOS Computational Biology

---

## [Editor Report · Acceptance letter]

11 Nov 2021

PCOMPBIOL-D-21-00590R2 

A nowcasting framework for correcting for reporting delays in malaria surveillance

Dear Dr Buckee,

I am pleased to inform you that your manuscript has been formally accepted for publication in PLOS Computational Biology. Your manuscript is now with our production department and you will be notified of the publication date in due course.

With kind regards,

Livia Horvath
